# Perfect absorption in complex scattering systems with or without hidden symmetries

Lei Chen [1,2 ✉], Tsampikos Kottos [3] & Steven M. Anlage [1,2 ✉]

Wavefront shaping (WFS) schemes for efficient energy deposition in weakly lossy targets is an ongoing challenge for many classical wave technologies relevant to next-generation telecommunications, long-range wireless power transfer, and electromagnetic warfare. In many circumstances these targets are embedded inside complicated enclosures which lack any type of (geometric or hidden) symmetry, such as complex networks, buildings, or vessels, where the hypersensitive nature of multiple interference paths challenges the viability of WFS protocols. We demonstrate the success of a general WFS scheme, based on coherent perfect absorption (CPA) electromagnetic protocols, by utilizing a network of coupled transmission lines with complex connectivity that enforces the absence of geometric symmetries. Our platform allows for control of the local losses inside the network and of the violation of time-reversal symmetry via a magnetic field; thus establishing CPA beyond its initial concept as the time-reversal of a laser cavity, while offering an opportunity for better insight into CPA formation via the implementation of semiclassical tools.

[1] Quantum Materials Center, Department of Physics, University of Maryland, College Park, MD 20742, USA. [2] Department of Electrical and Computer Engineering, University of Maryland, College Park, MD 20742, USA. [3] Wave Transport in Complex Systems Lab, Department of Physics, Wesleyan University, Middletown, CT 06459, USA. ✉email: LChen95@umd.edu; anlage@umd.edu

Coherent perfect absorption (CPA) has been appealing to physicists and engineers for both its fundamental and technological relevance. On the technological level, its implementation promises the realization of a family of wave-based devices performing highly-selective and tunable absorption in a manner that goes beyond the traditional concept of impedance matching. On the fundamental level, CPA has initially been associated with the concept of time-reversal (TR) symmetry, one of the most fundamental symmetries in nature. In its original conception CPA was proposed as the time reversal of a laser cavity[1,2]: specifically, it is a lossy cavity that acts as a perfect interferometric trap for incident radiation, provided that its spatial distribution matches the one that would be emitted from the same cavity if the loss mechanism is substituted by a corresponding gain mechanism i.e., if the cavity turns into a laser. Practically speaking, the CPA process works by injecting waves of particular amplitude and phase (coherent illumination)[2] from a number of input channels and causing them to interfere and to be completely absorbed by losses in the system. Remarkably, even an arbitrarily small amount of loss can be used to completely absorb the incident radiation if the system is sufficiently reverberant[3].

CPA phenomena have been theoretically proposed in a number of contributions[1,4–10], but only a few experimental works have reported a realization of CPA. At first, it was demonstrated with free-space counter-propagating waves impinging on lossy slabs in the form of semiconductors[2], metasurfaces[11], graphene-based structures[12], Parity-Time (PT) symmetric electronic circuits[13] and PT-symmetric quantum well waveguides that act as both a laser and CPA absorber[14], and acoustic systems[15,16]. Multi-port CPA was also achieved using a diffraction grating and lossy plasmonic modes (this work employed a pair of nonreciprocal scattering channels, but did not break time-reversal invariance (TRI))[17]. Most of these experimental demonstrations of CPA have generally been performed in open systems with freely counter-propagating waves arriving on a loss center at normal incidence. Such a configuration puts a strong constraint on the loss required to achieve CPA (e.g., 50% single-beam absorption[18]), and this is a significant limitation of such free space optical approaches. In summary, these early demonstrations used highly symmetric structures and excitation conditions to achieve CPA. Now the challenge is to considerably generalize the phenomenon and realize CPA in complex wave settings without special geometrical or hidden symmetries. It is clear that reverberations, hypersensitive complex interference, and system-specific characteristics (e.g., bouncing orbits in stadium billiards, coexistence of islands of regular dynamics in a chaotic sea of phase space of the underlying ray settings, mixed symmetries etc) blended with losses present in complex wave systems constitute a challenge for achieving CPA. Recently a demonstration of CPA was achieved in a multiple scattering environment with many input and output channels, implementing effectively a TR of a random laser[19]. This demonstration, however, utilized the conventional anti-laser concept and is limited to a mechanically-tunable loss element. It is desirable to expand the range of CPA to include complex and chaotic scattering environments of all kinds. Importantly, one has to investigate the applicability of CPA under controllable TR symmetry violation conditions.

In more precise terms, there are a number of deficiencies associated with the previous efforts to measure CPA. Some of these schemes failed to directly measure the outgoing waves from the system but deduced the CPA condition by calculating the output signals based upon combinations of data (usually the scattering matrix) taken under other (non-CPA) conditions. Obviously, a CPA platform that will allow for a direct measurement of the output signal will open up many technological opportunities, as proposed in the photonic context[3]. Secondly, the

degree to which the CPA condition is achieved has only been quantitatively demonstrated to a limited extent, typically 1 part in $10^2$, not at all close to the expected ideal outcome. Third, the previous experimental efforts have implemented loss in a way that is difficult to control and systematically vary, such as the thickness of a slab, or the temperature variation of conductivity. Finally, all previous works have been limited to systems that display TRI for the wave propagation (beyond the trivial TRI-breaking effects of dissipation).

Here we experimentally demonstrate the concept of CPA in a generalized setting where the weakly lossy cavity is a complex scattering system without any special geometric symmetries. We implement this scenario using a fully connected microwave network constructed from coaxial cables connected by Tee-junctions. By adding a convenient electronically tuned lossy attenuator, we can continuously and precisely control the nearly ideal CPA conditions, thus clearly identifying the CPA frequencies as the complex zeros of the scattering matrix which cross the real-frequency axis and achieving perfect absorption in this complex scattering setting. Most importantly, our experimental setup allows us to demonstrate that the concept of CPA can be extended beyond the case where TR symmetry holds, greatly expanding the impact and utility of the CPA phenomenon. The latter can be achieved by introducing a circulator into the microwave graph[20]. Such analysis proves that the concept of CPA goes far beyond its initial conception as a "time-reversed laser". Our experimental platform, due to its elegant simplicity, provides a convenient tool for the study of CPA in generic complex scattering systems having neither geometric nor dynamical symmetries. Importantly, it can be employed for the development of semiclassical schemes that utilize system-specific characteristics[21–23] aiming to the optimization of CPA traps. Finally, we have also confirmed the viability of CPAs in a two-dimensional quarter bow-tie chaotic billiard demonstrating beyond doubt that their formation occurs irrespective of the degree of complexity of the scattering process. Our results are general and apply to a variety of complex (i.e., without geometric or hidden symmetries) wave settings, ranging from optics and microwaves to acoustics and matter waves.

## Results

**Microwave networks.** Complex over-moded networks have been used to model mesoscopic quantum transport[24], electromagnetic energy propagation through multiply-connected arrays of compartments, and chains of coupled electromagnetic cavities[25]. In wave chaos studies[26–29], they have been proposed as a simple, yet powerful platform which under specific conditions[30–32] demonstrates all generic wave phenomena of systems with underlying classical chaotic dynamics. Their main advantage is that they allow for an exact semiclassical expansion while their wave scattering description is particularly transparent, inspiring the development and implementation of semiclassical[26–28,33] and super-symmetric[29–32,34] tools. Specifically, fully connected networks with incommensurate bond-lengths, under specific conditions[30–32], display universal statistical properties of various observables which are hypothesized to be described by random matrix theory (RMT)[35,36]. However, most practical systems also show deviations from universal chaotic behavior due to short orbits[37], mixed chaotic and regular phase space[38] (perhaps arising from parallel walls or soft boundaries), inhomogeneous loss, etc. In the case of fully connected networks with a small number of bonds (like the graph that we have used in our experiment, see Fig. 1) these deviations from RMT universality have already been identified in ref. [26] (see also ref. [33,39]) using semiclassics and their origin has been traced back to the presence of short periodic

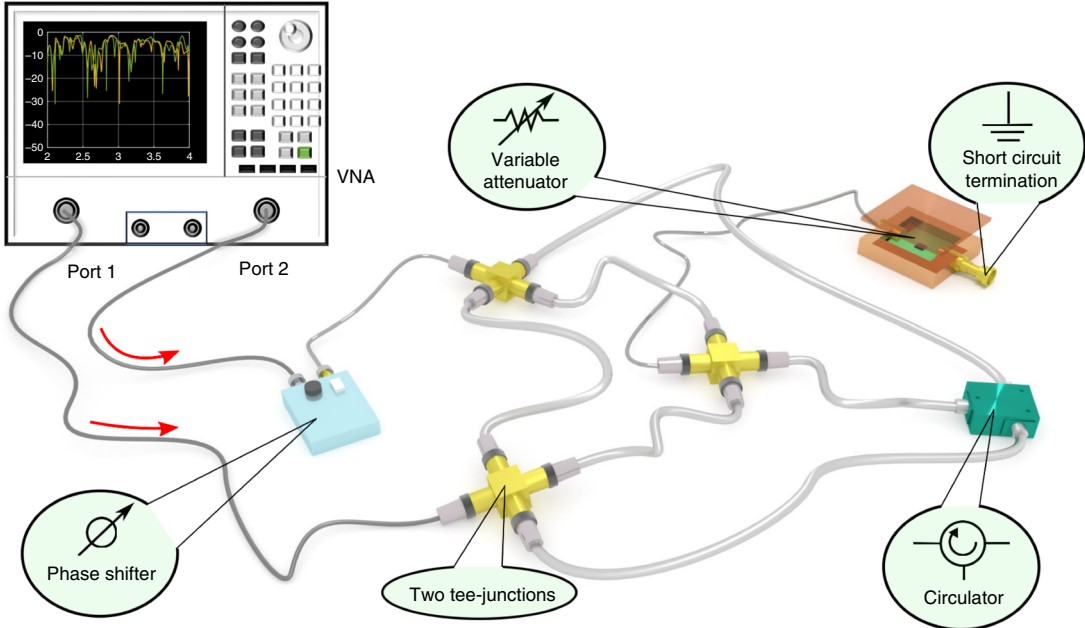

**Fig. 1 Experimental setup of the CPA state measurement.** A PNA-X (network analyzer with two internal sources) is used to generate microwaves with well-defined frequency and relative amplitudes at the two ports as the CPA state excitation signals. Coherent phase control between the two excitation signals is realized by placing a phase shifter between port 2 of the network analyzer and the graph. The outgoing and returning waves are directly measured by the PNA-X. On the right side of the figure is the tetrahedral microwave graph formed by coaxial cables and Tee-junctions. The four-way adapter shown in the figure is realized by connecting two Tee-junctions together in the real experiment. One node of the graph is loaded with a variable attenuator to provide parametric variation of the scattering system. One other node is made from either a Tee-junction (TRI) or a 3-port circulator (BTRI) to create a TRI system or a broken TRI system, respectively.

orbits that are trapped along individual bonds of the graph. Subsequent theoretical studies[28–32,34] have further established conditions under which RMT universality can be restored, while experimental implementations of graphs in the microwave realm[20,40–43] have provided additional evidence of the origin of these deviations[44–46]. Thus our platform, being a typical complex scattering system without any geometric symmetries and with controlled TR symmetries, and demonstrating both the extreme sensitivity to perturbations and typical deviations from universal statistical behavior[45]—due to system-specific features—is an ideal surrogate for testing the viability of CPA protocols in real-world scattering systems. Finally, to scrutinize further our statements on the feasibility of CPA implementation in complex systems (even in the case of chaos) we have performed additional experiments using a two-dimensional quarter bow-tie chaotic cavity[47].

Our experiment utilizes a tetrahedral microwave graph formed by coaxial cables and Tee-junctions[46,48]. A variable attenuator is attached to one internal node of the graph (see Fig. 1). The system is coupled to external transmission lines attached to $N$ specific nodes of the graph. In our specific setup, we utilize $N = 2$. Each coupling transmission line (labeled with a red arrow in Fig. 1) is a coaxial cable supporting a single propagating mode connecting to one port of the Vector Network Analyzer (VNA). The plane of calibration of the VNA is at the point where the transmission line attaches to the port of the graph. The experimental setup is completed with the addition of a phase shifter. The latter will be used in the second part of our experiments when we will launch the appropriate CPA waveforms into the complex network (see below).

**Analysis of the CPA state**. The wave transport properties of the microwave network are succinctly summarized by the $N \times N$ complex scattering matrix $S$. The latter connects the incoming and outgoing waves through these $N$ channels as $\phi_{out} = S\phi_{in}$,

where $\phi_{out}$ ($\phi_{in}$) is an $N$-component vector of outgoing (incoming) wave amplitudes and phases that defines the scattered outgoing (incoming) field in the channel-mode space. In the case of CPA all input energy is absorbed by the system, thus requiring $\phi_{out}$ to be zero. This physical condition is mathematically formulated by the requirement that $S\phi_{in} = 0$ for nonzero $\phi_{in}$. The latter condition is equivalent to the requirement that the $S$-matrix is not invertible i.e., it has a zero eigenvalue $\lambda_S = 0$. The associated eigenvector provides the incident waveform configuration that leads to a CPA. Note that this requirement does not violate any constraints of the $S$-matrix, which in the case of CPAs is sub-unitary due to the presence of an absorbing center inside the scattering domain. Let us finally point out that both the scattering matrix $S = S(\omega)$ and consequently its eigenvalues $\lambda_S = \lambda_S(\omega)$ are functions of the frequency $\omega$ of the incident waveform. From the mathematical perspective, one cannot exclude the possibility to have complex $\omega$'s as roots of the CPA condition $\lambda_S(\omega) = 0$. These complex zeros, however, are unphysical since they do not correspond to incident propagating plane waves and therefore have to be excluded from the set of acceptable CPA solutions. Of course, the reality of $\omega$ is not an issue in the experimental analysis since the measured $S$-matrix is always evaluated at real frequencies. From the above discussion, we deduce that a specific cavity (corresponding to a fixed connectivity, lengths of the cables of the graph, and loss strength) might support multiple CPAs i.e., different frequencies $\omega \neq \omega'$ for which the corresponding sub-unitary scattering matrices $S(\omega) \neq S(\omega')$ have a zero eigenvalue in their spectrum. We speculate that such multiple CPA scenarios will have higher probability to occur when the scattering matrices $S(\omega)$ and $S(\omega')$ are uncorrelated—a property that can be quantified by the rate with which the autocorrelation function $C(\chi) \equiv \mathcal{Re}\left\{\langle S_{\alpha,\beta}(\omega)S^*_{\alpha',\beta'}(\omega + \chi)\rangle\right\}$ goes to zero ($\langle \cdots \rangle$ indicates a spectral averaging). An interesting future research effort would be

to identify rigorous conditions under which such multiple CPAs can occur. We point out that a related analysis for the density of complex zeros of the S-matrix of a chaotic system has been recently carried out in ref. [10] (see also ref. [49]).

A straightforward way to determine experimentally the CPA conditions is via a direct evaluation of the eigenvalues {λ} of the measured S matrix and subsequent identification of the frequency ω for which the spectrum contains a zero. Such a direct process, however, is tedious and in many occasions, it turns out to be ineffective in our search for a true zero eigenvalue of the scattering matrix. Instead, we have utilized the parametric dependence of the S-matrix on the local attenuation strength in order to establish the zero eigenvalue condition. Specifically, we exploit simultaneously the frequency (wavelength) and local losses (attenuation) as two free parameters which allow us to exploit a larger parameter space for the identification of true S-matrix zeros. Once the CPA condition is satisfied, the required local loss and stimulus frequency are identified, and the corresponding S-matrix eigenvector which defines the CPA incoming stimulus wave amplitudes and phases (i.e., coherent excitation) is evaluated. The corresponding injected coherent monochromatic waveform results in a zero outgoing signal from all N scattering channels of the system. It should be noted that this procedure is entirely general and does not depend on the nature of the wave physics setting or on the degree of chaoticity that characterize the wave scattering process in the system.

Following this approach, the 2 × 2 scattering matrix of the graph is acquired using the setup of Fig. 1 (excluding the phase shifter). The measurement is taken from 10 MHz to 18 GHz which includes about 420 modes of the closed graph. The calibrated S-matrix of the 2-port graph is then measured under different attenuation settings ranging from 2 to 12 dB (which includes the insertion loss of the variable attenuator). Implementing a matrix diagonalization technique, the complex eigenvalues $\lambda_S$ of the S-matrix are found for each frequency and attenuator setting. A limited number of these eigenvalues are found to approach the origin in the complex $\lambda_S$ plane (see Fig. 2). These near-zero crossings are then examined in detail. Through this method, the specific frequencies and attenuation values at the

zero-crossing CPA state, as well as the required excitation relative magnitude and phase at the two ports (S-matrix eigenvector) are then determined.

Using the information obtained from the S-matrix measurement, the CPA conditions are identified, and we can directly test them experimentally. To do this, a two-source VNA is used to apply signals at the CPA frequency but with two different amplitudes (see Fig. 1). In addition, a phase shifter is added between port 2 of the network analyzer and the graph in order to deliver signals with the appropriate phase difference to the two ports of the graph. When signals are sent from both ports of the network analyzer simultaneously, it should be possible to observe the CPA, namely no microwave signals should emerge from the graph through either of the ports. The VNA measures both the outgoing and incoming waves at the plane of calibration, hence the CPA condition can be directly confirmed with this setup.

Under the CPA condition, a nearly perfect absorption is achieved, and it has been verified using four independent parametric sweep measurements (see Fig. 3). Both experimental and numerical data are plotted in the same figure. Parameters swept include the microwave frequency (wavelength), attenuation of the variable attenuator embedded in the graph, amplitude of excitation signal at port 1, and phase of excitation signal at port 2, while keeping other settings unchanged at the CPA condition. The input wave power and outgoing wave power are directly measured while changing the system configuration or the input stimulus setting. The ratio of outgoing signal power over input signal power ($P_{out}/P_{in}$) acquires values as low as $10^{-5}$ at the CPA condition, and both experiment and simulation show similar behavior upon deviation from the CPA conditions. Figure 3 demonstrates that the minimum outgoing power is measured at precisely the CPA condition, and rapidly increases in a cusp-like manner as any of the parameters deviate from that condition.

Due to the extreme sensitivity to perturbations and internal system details, it is naturally difficult to create a numerical model of a complex scattering system that reproduces all of its properties in detail. The numerical simulations in Figs. 2, 3, and 6 are based on S-parameter measurements of each individual component of the graph (Tee-junctions and coaxial cables) which are then

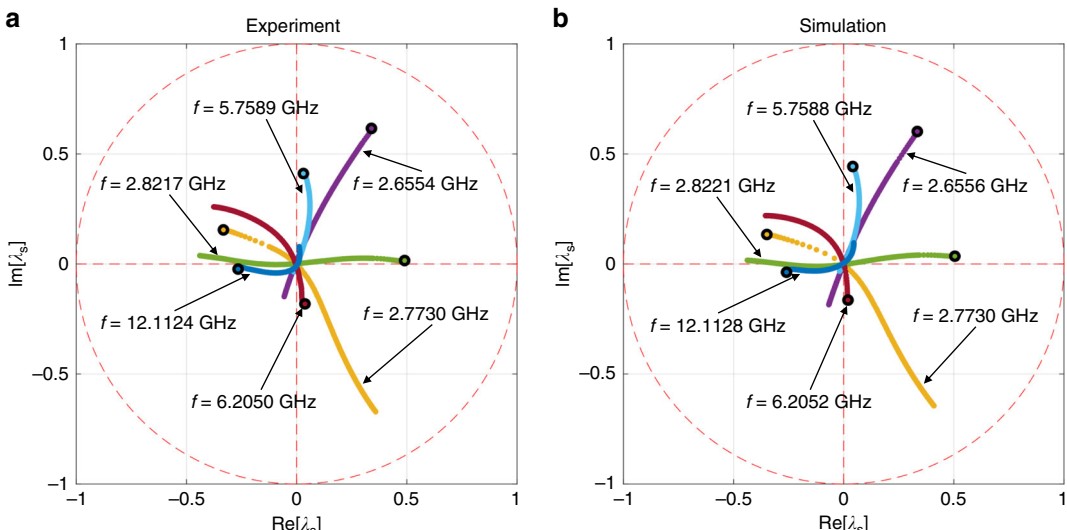

**Fig. 2 Plot of selected S-matrix eigenvalues as a function of attenuator setting in the tetrahedral graph showing examples of near-zero-crossing trajectories.** Selected eigenvalues of the S-matrix are plotted in the complex $\lambda_S$ plane, where the red dashed circle represents the unit circle. **a** Shows experimental data, while **b** shows data from the simulation. Each trajectory represents one frequency (color coded), and the corresponding frequency for each trace is labeled in the figure. The black circle at the start of every trajectory indicates the initial eigenvalue at minimum attenuation, and as attenuation increases, the eigenvalue goes nearly through the origin in the complex plane.

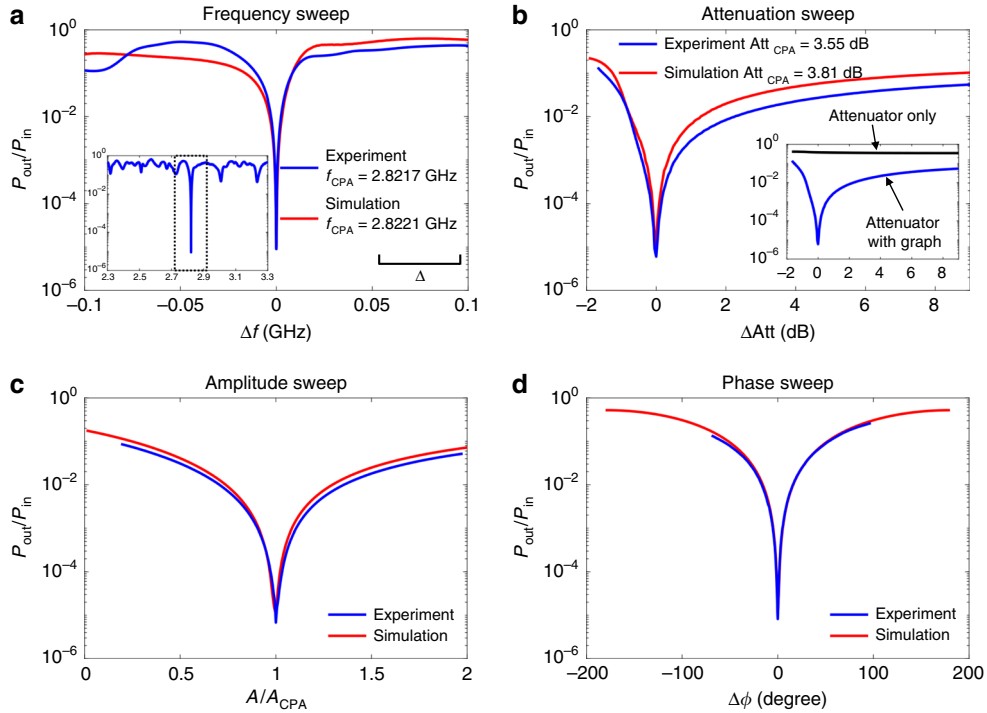

**Fig. 3 Evidence of CPA in the complex network under four independent parametric sweeps.** Plots are normalized so that CPA conditions are in the center of the parameter variation range. The closest frequency CPA condition for the simulation is plotted along with the experimental data. **a** Measured ratio of output power $P_{out}$ to input power $P_{in}$ as the microwave frequency sent into both ports of the graph is simultaneously swept near the CPA frequency ($\Delta f = f - f_{CPA}$). Inset shows the output-to-input power ratio response for a larger frequency range around the resonance, and the dashed box corresponds to the frequency range shown in **a**. The output-to-input power ratio shows a sharp dip close to $10^{-5}$ at the CPA frequency ($f_{CPA}$) in both experiment and simulation. The scale bar of the mean mode spacing $\Delta$ is shown in the plot for reference. **b** Output-to-input power ratio obtained by varying the attenuation of the variable attenuator in the graph, while the other waveform characteristics (CPA frequency and waveform) are equal to the ones set in **a**. $\Delta$Att is the attenuation normalized by Att$_{CPA}$ from the CPA condition. Inset shows the absorption difference between the attenuator only and the attenuator embedded in the graph. Output-to-input power ratio obtained by changing the amplitude $A$ (**c**) and phase difference $\Delta\phi$ (**d**) separately of the two excitation signals required for the CPA state. The absorption of power reaches its maximum at the CPA configuration, and quickly deteriorates for even small offset from the CPA condition. All experimental results are obtained by direct measurement of the input and output RF powers.

combined with the same topology as the full graph to yield high-fidelity descriptions of the data (see Supplementary Note 3). It should be noted that a numerical model of the graph employing idealized components (e.g., without taking into account the frequency dependence of their impedance) also shows the CPA conditions, although at different frequencies and attenuator settings. The models demonstrate that the CPA results are generic to complex scattering systems, establishing the breadth and generality of our results.

To emphasize the importance of having a reverberant cavity instead of a bare attenuator only, we measure the power ratio of the bare attenuator (see Fig. 3b inset) under the same settings as in the complex networks. From the inset, we can see that in the absence of the graph, the attenuator can only absorb a small fraction of the incident power ($P_{out}/P_{in} > 10^{-1}$). This illustrates the importance of having the complex network as the cavity to create the CPA condition.

Both the variable attenuator and the microwave graph cavity play important roles in the formation of CPAs. At the same time, in realistic settings there are other elements that might also contribute to the total absorption. To rule out their influence in the CPA protocol, we have evaluated their contribution to the total power absorption using the idealized simulation model shown in Fig. 4a (for further details see the Methods section). Figure 4b shows that the voltage amplitudes at the four nodes in the graph under CPA condition are roughly equal. As shown in Fig. 4c, most of the power (i.e., more than 80%) is absorbed by

the variable attenuator, and the rest is absorbed by the coaxial cables, which contribute to a spatially uniform absorption inside the system. There is very little reactive power in the graph under the CPA condition, as opposed to the "Anti-CPA" state where a large amount of reactive power circulates in the system (see Supplementary Fig. 2c). Therefore, Figure 4 exactly demonstrates what the theory predicts: the CPA is the combined effect of localized loss and intricate wave interferences, providing a perfect destructive interferometric trap for the incident radiation. The importance of these specific interferences that are induced via the above CPA protocol, and its dominance over other (nonuniversal) effects is even more appreciated in the case of our tetrahedral graph. Here, short periodic orbits associated with an enhanced backscattering at the vertices promote a trapping of the electromagnetic field in individual cables of the graph (i.e., a scarring effect[31–33,45]) which might not include the lossy element (attenuator). Therefore, one could argue that their presence poses fundamental difficulties for the realization of CPAs due to a localized lossy center which is placed somewhere else inside the cavity. Our experimental results demonstrate beyond doubt that the interference imposed via the CPA protocol prevails over all these nonuniversal features, leading to a (almost) perfect absorption of the coherent incident radiation. Since the existence of nonuniversal features of various origin is typical in any realistic complex system, we expect that the development of a semiclassical theory of CPA (which utilize nonuniversal features), will lead to a better design of optimal traps. Complex

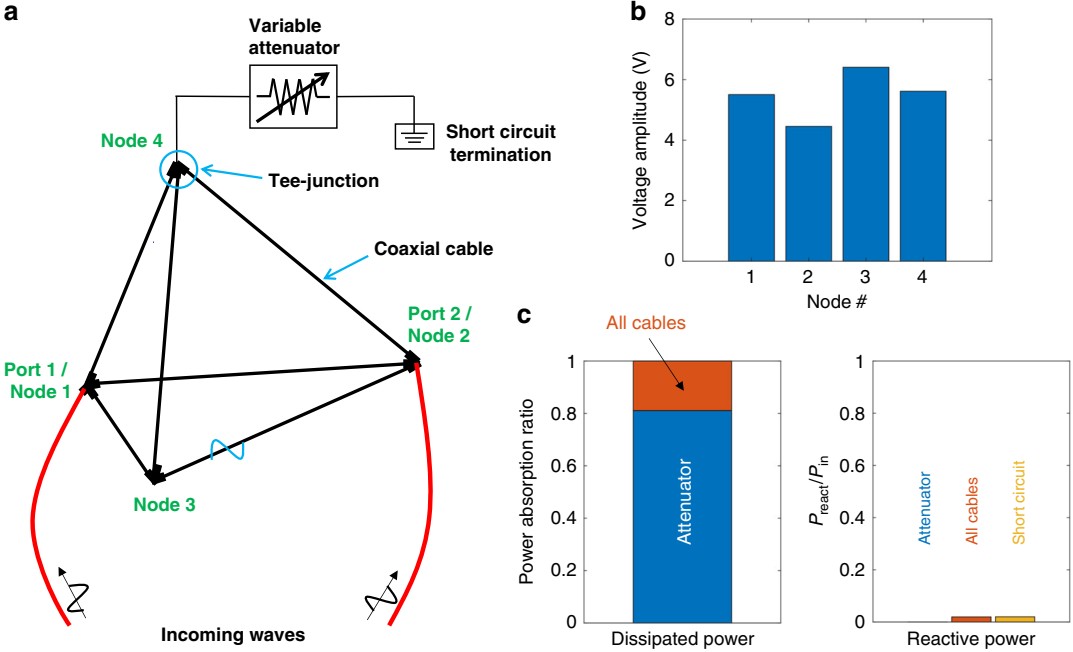

**Fig. 4 Voltage profile and power distribution of CPA state in idealized simulation. a** Schematic of the microwave graph with labeled ports under CPA condition at 2.2999 GHz in simulation. **b** Voltage profiles of four nodes in the graph under CPA condition. **c** Power distribution among the graph components under the CPA condition. Left plot shows that about 80% of the power are being dissipated on the attenuator, while the remainder is dissipated in the uniformly attenuated cables. Right plot shows reactive power on the cables and short circuit. Compare with the "Anti-CPA" condition in Supplementary Fig. 2.

networks can offer a fertile platform for developing and testing such theories.

**CPA in 2D chaotic quarter bow-tie billiard.** To further challenge the robustness of the CPA protocols, we have also implemented them using a two-dimensional quarter bow-tie cavity, shown in the inset of Fig. 5. Such cavities are known to demonstrate chaotic dynamics in the classical (ray) limit and have been used in the past as an archetype system for wave chaos studies[47,50–52]. The local losses have been incorporated via the same voltage variable attenuator (see details in the "Methods" section) which is attached to a port at the red dot position in the schematic. There are two additional coupling ports on the top plate of the bow-tie billiard for measurements. Following the same experimental procedure as previously discussed, we have identified the CPA conditions and injected the corresponding CPA waveform into the cavity. Due to the higher mode density in two-dimensional billiards, an interesting feature is the appearance of two zeros $\lambda(\omega_1) = \lambda(\omega_2) = 0$ at the same attenuation strength but two different frequencies. At these frequencies (keeping the attenuation parameter fixed), the system supports two different CPA waveforms identified by two different eigenvectors of the $S$-matrix. We have confirmed this statement via a direct injection of these specific waveforms into the cavity and measuring the corresponding output power versus frequency for a fixed attenuation (see Fig. 5). At the CPA frequencies, we find that the output power associated with these two distinct waveforms drops sharply as one expects from a CPA. Therefore, a CPA setup can be utilized as a fast tunable switch where incident monochromatic radiation from one port of the cavity is interferometrically suppressed by a control signal that is injected from the other port.

**Extending CPA beyond time-reversal invariance.** After exploring the formation of a CPA in a TRI tetrahedral microwave

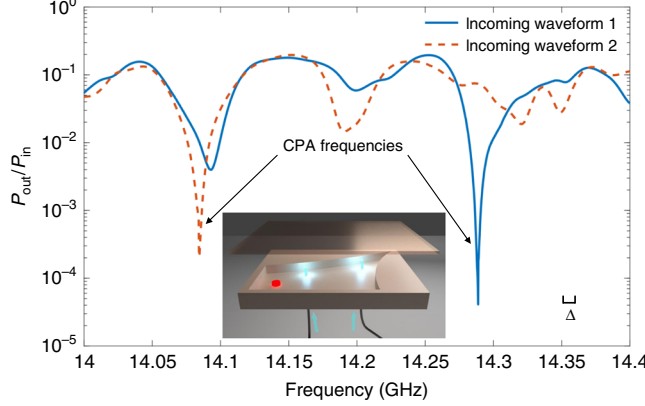

**Fig. 5 Demonstration of two CPA states in the quarter bow-tie billiard.** A two-dimensional quarter bow-tie billiard (see inset) is used to test the CPA protocols. The red dot on the bow-tie represents the location of a point-like variable loss in the cavity. Through analysis of the $S$-matrix, two CPA states are found at the same system configuration. By injecting two different CPA waveforms, the measured ratios of output power $P_{out}$ to input power $P_{in}$ as a function of the microwave frequency are plotted together. The different incoming waveforms support two different CPA states with different CPA frequencies. The scale bar of the mean mode spacing $\Delta$ is shown in the plot as well.

graph, we turned our focus to a graph with BTRI (Broken-Time Reversal Invariance). CPA associated with violated TR symmetry is unconventional, and challenges the idea that CPA is simply a time-reversed laser action[1,2]. Driven by such motivation, we introduced a circulator[20] (2–4 GHz) at one internal node of the tetrahedral graph (see Fig. 1), which allows us to violate the TRI in a controllable manner. Previous work demonstrated that the statistics of the microwave graph impedance (or reaction matrix)

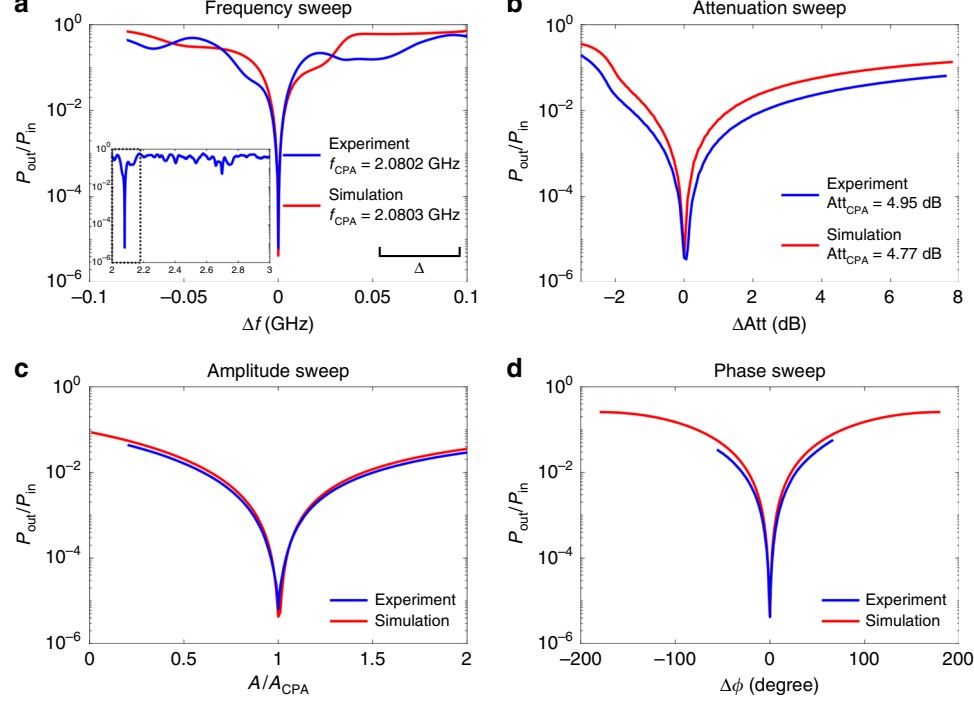

**Fig. 6 Evidence of CPA in the complex network with BTRI under four independent parametric sweeps.** Plots are normalized so that CPA conditions are in the center of the parameter variation range. The closest CPA frequency condition for the simulation is plotted along with the experimental data. **a** Measured ratio of output power $P_{out}$ to input power $P_{in}$ as the microwave frequency sent into both ports of the graph is simultaneously swept near the CPA frequency ($\Delta f = f - f_{CPA}$). Inset shows the output-to-input power ratio response for a larger frequency range around the resonance, and the dashed box corresponds to the frequency range shown in **a**. The output-to-input power ratio shows a sharp dip below $10^{-5}$ at the CPA frequency ($f_{CPA}$) in both experiment and simulation. The scale bar of the mean mode spacing $\Delta$ is shown in the plot for reference. **b** Output-to-input power ratio obtained by varying the attenuation of the variable attenuator in the graph, while the other waveform characteristics (CPA frequency and waveform) are equal to the ones set in **a**. $\Delta$Att is the attenuation normalized by Att$_{CPA}$ from the CPA condition. Output-to-input power ratio obtained by changing the amplitude $A$ (**c**) and phase difference $\Delta\phi$ (**d**) separately of the two excitation signals required for the CPA state. All experimental results are obtained by direct measurement of the input and output RF powers.

changed from that characterized by the Gaussian orthogonal ensemble of random matrices (appropriate for TRI systems) to the Gaussian unitary ensemble (appropriate for BTRI systems) with the addition of this circulator[20,53,54].

Following the same procedure as for the TRI graph experiment, the CPA conditions are found by evaluating the eigenvalues of the S-matrix. After that, similar sweep measurements are done as in the TRI case to directly verify the formation of a CPA in the BTRI graph. Our experimental measurements are reported in Fig. 6 and confirm the formation of a CPA despite the naive expectation that the presence of a nonreciprocal element (circulator) in the system should weaken the coherence between incident waves, as seen in the eigenfunctions of BTRI wave chaotic systems[55]. The simulations from our modeling are reported in the same figure and show the same behavior as the experimental data. The formed CPAs show the same characteristic features (e.g., sharp resonance, sensitivity to various parameters, abrupt drop of outgoing signal) as the ones reported in the TRI case. We therefore conclude that the CPA protocol applies even to BTRI systems.

## Discussion

The implementation of CPA in generic complex scattering systems opens up a number of interesting applications beyond the ones that we have already discussed (e.g., reconfigurable switching). The first is long-range wireless electromagnetic power transfer technologies that seek to deliver significant electromagnetic power to a single designated object located inside a complex scattering enclosure many wavelengths away from the source. Current approaches utilize multiple scattering and interfering wave trajectories connecting power source and target through either TR[56,57] or phase conjugation of microwave signals[58] that involve either large bandwidth or a large numbers of channels. These methods suffer from low efficiency as well as radiation safety concerns. A CPA-based method would require only that the target employs a tunable loss, or other tunable scattering property[59–61], and that the source employs only a small number of channels to measure the S-matrix of the enclosure to find the CPA condition. Once CPA is established, the source would output a coherent energetic signal that would be maximally absorbed at the desired target, with minimal loss elsewhere in the environment. As an added benefit other users can utilize the same bandwidth to perform other tasks (e.g., information transfer) utilizing the "Anti-CPA" condition (see Supplementary Note 2), alleviating frequency crowding concerns.

A second application concerns sensing of minute changes in a scattering environment. There will be a sensitive change in absorbed energy, or output power from a complex scattering structure, due to any perturbation of the system from the CPA condition, as illustrated by the cusp-like features in Figs. 3, 5, and 6. This arises from the shift of the scattering matrix zero off of the real-frequency-axis, and the dramatic alteration of the scattering matrix is a very direct and easy to measure property. This sensing protocol is simpler than the frequency splitting of degenerate modes created by a perturbation to an optical resonator tuned to

an exceptional point[62], for example. Our CPA approach, which relies on the natural complexity of the cavity, is generic and will work in any wavelength regime (or for any wave phenomenon) as long as it is performed in a complex scattering environment.

A third example application is a CPA protocol for secure communications. Consider that the absorber is a target receiver embedded at an unknown location in a complex environment. Due to the complexity of the environment (multipaths with sensitive interference), transfer of information from an outside source occurs only if the emitter prepares and injects a very specific waveform. The waveform has to be determined by the absorber property as well as the environment information, and is rapidly altered as soon as the absorber changes its property. The CPA conditions (absorption and frequency) could be utilized to create a unique key to encrypt the communication and secure the transmission process. Through this method one can establish a secure communication protocol between the emitter and the absorber. A related application is to utilize CPA as a switch for an arbitrary incident signal at one frequency. For a given waveform incident through the ports of the system one can arrange the relative amplitude/phase of a control wave injected into the CPA cavity to create complete absorption of an incident signal at the same frequency. Switching back and forth between the CPA and anti-CPA control waves will toggle the incident signal to a maximum extent.

In summary, we demonstrate the implementation of CPA protocols in generic complex scattering systems without any geometric or hidden symmetries. The primary platform that has been used in our investigations was a microwave realization of a quantum graph consisting of a complex network of coaxial cables where TR symmetry can be preserved or violated in a controllable manner. Irrespective of the symmetry, the CPA condition has been realized through continuous tuning of a localized lossy component and its efficiency has been tested by means of direct measurement of RF power coming out of the graph. As much as 99.999% of the injected power is absorbed by the system. To get additional confirmation of the efficacy of our experimental CPA protocols in complex systems, we have also tested them successfully in a chaotic microwave bow-tie cavity. Our work demonstrates that CPA can indeed be achieved even in the case of complex (or chaotic) scattering setups where small variations in the form of the incoming waves or of the scattering system might lead to dramatic changes in the scattering fields. These findings establish the validity of CPA protocols, independent of the degree of complexity of the wave transport phenomena, originating either from the influence of system-specific features in the scattering process or from the presence or the absence of an underlying classical chaotic dynamics. Importantly, our work generalizes the operations and settings for CPA beyond its initial assumptions of TR symmetry and is expected to motivate practical applications, including designing efficient absorbers, sensitive reconfigurable switches, enabling practical long-range wireless power transfer, and associated high-efficiency energy conversion systems. The extreme sensitivity of absorption to parametric variation away from the CPA condition can be utilized for ultra-sensitive detectors and secure communication links. These ideas translate to all forms of complex wave scattering, including audio acoustics and solid-body vibro-acoustics. For future work, the CPA phenomenon can be extended to the nonlinear regime[63] by introducing nonlinear elements into the system.

## Methods

**Experimental setup**. Our main experimental setup is a tetrahedral microwave graph constructed from six coaxial cables connected by coaxial Tee-junctions. The cables are semiflexible SF-141 coaxial cables, each of different length, with SMA male connectors on both ends (Model SCA49141) obtained from Fairview

Microwave, Inc. The dielectric material of the cable is solid polytetrafluoroethylene, which has a relative dielectric constant of 2.1. The inner conductor of the cable is silver plated copper clad steel, and has a diameter of 0.036 in. (0.92 mm); while the outer shield is a copper–tin composite which has an inner diameter of 0.117 in. (2.98 mm). The dielectric loss tangent of the medium is $tan\delta = 0.00028$ at 3 GHz, and the resistivity of the metals in the cable is $\rho = 4.4 \times 10^{-8}$ $\Omega \cdot$m at 20 °C. Both of these contribute to the uniform loss of the coaxial cables. The lengths of the six cables are 13, 14, 15, 16, 18, and 20 in. The total length of the graph is then ~2.44 m, giving rise to a mean spacing between modes of 42.4 MHz, which is constant as a function of frequency. On one node of the graph, two Tee-junctions form a four-way adapter where a voltage variable attenuator (HMC346ALC3B from Analog Devices, Inc.) is connected to one connector. A short circuit termination is connected to the other end of the attenuator. Using a Keithley power supply (2231A-30-3), the attenuation of the variable attenuator is continuously swept by varying the supplied voltage from 4.00 to 7.00 V. To find the appropriate CPA condition of the setup, we perform the S-matrix measurement of the graph (using the PNA-X N5242A from Agilent Technologies, Inc.) in the frequency range from 10 MHz to 18.01 GHz (at 96,001 equidistant frequency points) which includes about 420 modes of the closed graph, with varying attenuation from about 2 to 12 dB (which includes the insertion loss of the variable attenuator). The attenuation is swept with a step size of roughly 0.1 dB. In the case of a BTRI microwave graph, a ferrite circulator (Model CT-3042-O from UTE Microwave Inc.) is added to one node of the graph (see Fig. 1). The circulator has an operational frequency range from 2 to 4 GHz, which constrains the frequency range of measurement accordingly. By connecting the microwave graph to a 2-port VNA, with a coupling strength of about 0.68, we can obtain the S-matrix of the system under different attenuation configurations.

The quarter bow-tie billiard, shown schematically in Fig. 5, has an area of $A = 0.115$ m². The brass cavity has a horizontal length of 17.0 in. (43.2 cm), and a vertical length of 8.5 in. (21.6 cm). The upper arc radius is 42.0 in. (106.68 cm), and the right arc has a radius of 25.5 in. (64.8 cm). The height of the cavity is $d = 0.3125$ in. (7.9 mm), which makes it a quasi-2D billiard below the cutoff frequency of $f_{max} = c/(2d) = 18.9$ GHz. We add the voltage variable attenuator to the top plate of the billiard by means of a coaxial port at the red dot location (see inset of Fig. 5) as the local loss. A stub tuner (1819D from Maury Microwave Corporation) is used to tune the coupling between the variable attenuator and the cavity.

**Verification of the CPA state**. In order to create the coherent stimulus signals, we use a two-source VNA (PNA-X N5242A from Agilent Technologies, Inc.) to serve as the RF signal source and measure the incoming and outgoing wave energies as well. The PNA-X has two built-in RF sources which provides great convenience for us to individually adjust the amplitudes of the two input excitation signals. The relative phase difference of the two input signals is controlled by adding a manual coaxial phase shifter (Model 3753B from L3Harris Narda-MITEQ) between the VNA and port 2 of the graph (see Fig. 1). With this measurement setup, we can effectively tune the input stimulus signals for the CPA state as well as the system configurations, perform comprehensive parametric sweep measurements (see Figs. 3, 5, and 6), and directly measure the input power $P_{in}$ and output power $P_{out}$ of the graph.

**Simulation model**. To compare with the experimental results, we set up a comparable simulation model in CST (Computer Simulation Technology) Studio. CST is commercial software specifically designed for electromagnetic field simulation, and we use the Circuits and Systems module to simulate the microwave graph. All individual components in the graph experiment, e.g., coaxial cables and Tee-junctions, are modeled by their measured S-matrix data at the exact same frequency points used in the experiment. The S-matrix data for the variable attenuator are measured at the designated supply voltages from 4.00 to 7.00 V. The S-matrix data are imported as TOUCHSTONE file blocks in the simulation model, and correctly capture the electrical characteristics of all components. The imported S-matrices are then combined in the same topology as the graph of interest. Therefore, following the same procedure as in the experiment, we can verify the CPA phenomena in the simulation as well.

In order to better understand the power distribution inside the system under CPA conditions, we adapt an idealized simulation model in CST. In this model, nodes constructed from Tee-junctions are set to be ideal (no loss), and the attenuator is set to have no frequency-dependent characteristics, and the coaxial cables have uniform attenuation properties. Results are shown in Fig. 4 and Supplementary Fig. 2.

## Data availability
The data that support the findings of this study are available in the Digital Repository at the University of Maryland (DRUM) with the identifier "doi:10.13016/aqny-7v5z" [http://hdl.handle.net/1903/26379][64].

## Code availability
The custom codes that produce results presented in this paper and other findings of this study are available from the corresponding authors upon reasonable request.

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

## Acknowledgements

This work is supported by the AFOSR under COE Grant FA9550-15-10171, the ONR under Grants N000141912481 and N00014-19-1-2480, and the Maryland Quantum Materials Center. We acknowledge S. Suwunnarat for helping us with the drawing of Fig. 1. Partial funding for open access provided by the UMD Libraries' Open Access Publishing Fund.

## Author contributions

L.C. conducted the measurements and carried out the simulation under the supervision of S.M.A. L.C. performed the data analysis under the guidance of T.K. and S.M.A. L.C. wrote the manuscript with input from all co-authors.

## Competing interests

The authors declare no competing interests.
