## [Peer Review File · Nature Communications]

Reviewers' comments:

Reviewer #1 (Remarks to the Author):

In the manuscript with the title 'Perfect absorption in chaotic cavities with or without hidden symmetries' coherent perfect absorption is achieved in microwave networks, that is, an experimental realization of quantum graphs, also in microwave networks with violated time-reversal invariance. The manuscript deserves publication, however, not in Nature Communications.

This is just another realization of a CPA in a microwave network, which in the ideal case is easier to control than, e.g., the system used in Ref. [22].

Still, while applicability to real systems is straightforward in the latter system, this in my opinion is not the case for microwave networks. The authors already claim in the abstract that their realization opens up CPA to many more applications. They should explain explicitly in what kind of systems and how it can be implemented.

It is well known, that quantum graphs / microwave networks have some drawbacks concerning the realization of fully chaotic systems. As already mentioned in a publication by one of the authors, Tsampikos Kottos, and U. Smilansky [Ann. Phys. 274, 76 (1999)] and outlined in a recent experimental work [Dietz, Yunko, Bialous, Bauch, Lawniczak and Sirko, PRE 95, 052202 (2017)] the spectral properties of such networks exhibit deviations from those of fully chaotic systems. These may be attributed to backscattering at the vertices of the network, which leads to the occurrence of short orbits, which bounce back and forth on individual bonds. I would expect that such orbits hinder the realization of perfect CPA conditions. The authors should comment on this and explain in more detail how they were able to achieve 99.999 % absorption!

Furthermore, quantum graphs are defined by their vertex scattering matrix. However, Tee-joints and especially the combination of two Tee-joints to form a 4 joint, fulfill the required scattering features only approximately. Accordingly, the (semiclassical) theories derived for ideal quantum graphs do not necessarily apply to networks. Again a comment on this imperfectness and the impact on the CPA realization should be provided by the authors.

Reviewer #2 (Remarks to the Author):

Coherent perfect absorption (CPA) process uses interference of waves coming to the system from different channels. The resulting waves should be completely absorbed by losses in the system.

In the manuscript CPA is realized using microwave networks with and without time reversal symmetry. The choice of the systems is well thought because microwave networks as the only ones allow for the experimental simulation of quantum systems corresponding to all three classical ensembles (GOE, GUE and GSE) in the random matrix theory.

The microwave experiments presented in the manuscript demonstrate very clearly the action of CPA, both for the networks with time reversal symmetry as well as for the ones with broken time reversal symmetry. This is the first CPA implementation in a system with broken time reversal symmetry. The key ingredient of the successful demonstration of CPA is the use of VNA with two internal sources which can contribute to forming interference phenomena in CPA. Convincing experimental results are additionally complemented by the theoretical ones.

Some remarks:

The title of the manuscript "Perfect Absorption in Chaotic Cavities with or without Hidden Symmetries" is somewhat misleading because the authors deal rather with perfect absorption in networks.

Microwave networks with broken time reversal symmetry which utilized microwave circulators were introduced in Phys. Rev. E 81, 046204 (2010). This reference should be cited additionally to the references [27] and [28].

Summarizing, the manuscript is well written and reports important new results. I recommend it for the publication.

Reviewer #3 (Remarks to the Author):

The presented manuscript reports experimental investigation on the realization of an single frequency absorber in a microwave junction network. The approach used is based on ideas from coherent perfect absorption (CPA) in complex systems. The main claim of the paper, as stated in the introduction, is to have achieved CPA in a classical chaotic structure without symmetries.

I am not convinced of the validity of this paper, not of its merit for NCOMM. The list of my comments is as follows:

- 1) The abstract of the paper focuses on biological applications, however, in the paper nothing is discussed except the realization of a single frequency absorber in a microwave system. On the theme of perfect absorption there is a vast literature in photonics made by hundreds of papers that exploits many different approaches and not limited by the CPA. Providing a work on an absorber, not even at photonics frequencies but at microwaves, without demonstrating any important application, does not certainly met the high standards of NCOMM but just deserve specialized literature.
- 2) I do not agree the structure of the authors is chaotic. On the contrary, is a simple network made by 4 segment and there is no chaos in it. As a proof of this statement, the paper lack any sort of chaotic characterization of the system, e.g., the KS entropy, the Lyapunov exponent,... or any other metric that can demonstrate that the system is chaotic. The system used it is simply a network made by four segment.
- 3) Besides these points, I want to emphasize that this work, in a nutshell, is based on adjusting the parameters of an elementary microwave network to achieve an absorbing state at a specific frequency. That is to say, the while work is about engineering a simple filter, with no application described. In this respect, this is simple optimization work that should be published in specialized journals.
- 4) As a minor note, I also observe a huge discrepancy between simulations and theory (Fig. 2): all predictions are different and do not follow the same evolution. It is not clear to me what the authors want to demonstrate with this analysis: it just shows a very confusing description.

In summary, I find this paper very weak. Its merit is not properly addressed, its scope is very narrow and the results described are a trivial optimization of a simple microwave network. I suggest the authors to submit this work to a more specialized paper

NCOMM CPA Rebuttal Letter

We thank the referees for very helpful critical remarks. We believe that our work has not been properly understood by the referees, and this is partly due to the way in which we presented our work.

Here are the general issues that we have addressed in the revised manuscript. First is the role of chaos. Chaos is not required in our experimental system to achieve CPA (it is also not a hindrance). In addition, the extent to which our experiment can be considered chaotic, the extent to which it fulfills the assumptions of semiclassical theory, or the extent to which our simulations match the data, are all irrelevant to our CPA results and the importance and generality of our conclusions. Indeed the tetrahedral graph shows some statistical properties that are consistent with certain universal expectations of wave chaotic systems based on random matrix theory, but it also shows deviations, but this is very much like *all* practical realizations of wave chaotic systems. Nevertheless, with or without chaos, the CPA condition can be satisfied, and it is this generality that makes our results so important. Our point (better emphasized now) is that the extreme sensitivity to perturbations implied by underlying chaos does not prevent us from achieving CPA, and we now state this explicitly. The second issue is the role of applications, and we have now addressed this more thoroughly in the manuscript. Two of the referees imply that identifying new applications is a requirement for publication in Nature Communications. We discuss applications more explicitly and fully now. Overall, we argue more carefully and convincingly about the generality and novelty of our results, and try to guide the reader to see that the results are not limited to microwave networks, or in any way by the presence or absence of chaos.

Here is the detailed response to the referee comments.

Reviewer #1

This is just another realization of a CPA in a microwave network, which in the ideal case is easier to control than, e.g., the system used in Ref. [22] {Pichler, et al., now Ref. [23]}. Still, while applicability to real systems is straightforward in the latter system, this in my opinion is not the case for microwave networks. The authors already claim in the abstract that their realization opens up CPA to many more applications. They should explain explicitly in what kind of systems and how it can be implemented.

This demonstration of CPA is uniquely different from all previous demonstrations in that it is accomplished in a generic complex system that lacks time-reversal symmetry, used in all previous demonstrations. It works with or without chaos, or with any degree of “imperfect chaos”, and is not restricted to just microwave networks. In fact, the complexity in the scattering is due to the network connectivity which enforces (in the classical limit) folding and stretching – the two elements required for Bernoulli dynamics. A few applications were mentioned in the original text, but we now elaborate more extensively on three clear novel applications of our CPA demonstration in the penultimate paragraph (lines 231-264) of the main text.

It is well known, that quantum graphs / microwave networks have some drawbacks concerning the realization of fully chaotic systems. As already mentioned in a publication by one of the authors, Tsampikos Kottos, and U. Smilansky [Ann. Phys. 274, 76 (1999)] and outlined in a recent experimental work [Dietz, Yunko, Bialous, Bauch, Lawniczak and Sirko, PRE 95, 052202 (2017)] the spectral properties of such networks exhibit deviations from those of fully chaotic systems. These may be attributed to backscattering at the vertices of the network, which leads to the occurrence of short orbits, which bounce back and forth on individual bonds. I would expect that such orbits hinder the realization of perfect CPA conditions. The authors should comment on this and explain in more detail how they were able to achieve 99.999 % absorption!

Upon reading these remarks we realized that a reader might get the impression from our paper that full chaos (in the classical limit) is necessary for our implementation of CPA. In fact it is not. We now explicitly state that chaos is not a requirement in the fourth paragraph and conclusion paragraph of the text. Indeed the tetrahedral graph has some properties characteristic of wave chaotic systems (as described by Random Matrix Theory) and other properties that are not. In fact this illustrates a key strength of our approach. Almost all practical realizations of “chaotic” wave systems suffer from some degree of “non-chaotic” behavior. Our approach to CPA works for all of these systems, no matter what detailed properties shows deviations from the predictions of random matrix theory. We re-iterate that our approach is broad and generic and does not require any peculiar properties of the system to achieve “perfect CPA conditions”. In fact, this is one of the main messages that our work is establishing. We very much appreciate this comment and now clearly state these points in the abstract, paragraph 4 (lines 99-112), and the conclusion paragraph. We also now cite the paper by Dietz, *et al.* {new ref. [29]} mentioned by the referee as part of our strengthened arguments. Finally, we wish to emphasize that despite the sensitivity that is inherent to chaotic/complex scattering, our experiment demonstrates that CPA can still be achieved.

Furthermore, quantum graphs are defined by their vertex scattering matrix. However, Tee-joints and especially the combination of two Tee-joints to form a 4 joint, fulfill the required scattering features only approximately. Accordingly, the (semiclassical) theories derived for ideal quantum graphs do not necessarily apply to networks. Again a comment on this imperfectness and the impact on the CPA realization should be provided by the authors.

Again this comment is an opportunity to point out the generic strength and utility of our approach to achieving CPA. Our method is insensitive to the relevance (or lack thereof) of semiclassical theoretical approaches to describe the system—though we maintain that our platform is indeed susceptible to a semiclassical treatment; the only required input is knowledge of the vertex scattering matrix that can be measured experimentally. {As an aside, semiclassical theories have been successfully applied for many different types of vertex scattering matrices (For a generalization to abstract vertex scattering matrices σ see for example: (a) G.

Berkolaiko, H. Schanz, R. S. Whitney, Phys. Rev. Lett. 88, 104101 (2002); (b) G. Tanner, J. Phys. A: Math. Gen. 34, 8485 (2001); Z. Pluhar, H. A. Weidenmuller, Phys. Rev. Lett. 112, 144102 (2014).). For example, in Kottos and Smilansky. Ann. Phys. 274, 76 (1999) the authors considered, within a semiclassical formalism, a scenario where the vertex scattering matrices are frequency dependent due to mixed boundary conditions at the vertices.} But once again, the degree of complexity/chaoticity of the scattering process is irrelevant to the implementation of the CPA approach. In fact, our experiment demonstrated exactly this: that despite the complexity of the scattering process, we still can realize a CPA.

Reviewer #2:

Coherent perfect absorption (CPA) process uses inference of waves coming to the system from different channels. The resulting waves should be completely absorbed by losses in the system. In the manuscript CPA is realized using microwave networks with and without time reversal symmetry. The choice of the systems is well thought because microwave networks as the only ones allow for the experimental simulation of quantum systems corresponding to all three classical ensembles (GOE,GUE and GSE) in the random matrix theory. The microwave experiments presented in the manuscript demonstrate very clearly the action of CPA, both for the networks with time reversal symmetry as well as for the ones with broken time reversal symmetry. This is the first CPA implementation in a system with broken time reversal symmetry. The key ingredient of the successful demonstration of CPA is the use of VNA with two internal sources which can contribute to forming interference phenomena in CPA. Convincing experimental results are additionally complemented by the theoretical ones.

We thank the reviewer for succinctly summarizing some important points: The implementation of CPA in the framework of complex/chaotic scattering is a unique contribution of the present work. Our experimental platform is unique in that it can give access to the three main university classes of generic complex scattering systems. This is the first demonstration of CPA in time-reversal broken systems, throwing the traditional interpretation of CPA as a time-reversed laser completely on its head. And the experimental and theoretical results demonstrating CPA are appropriate for, and at the accepted standards of, those expected from complex scattering networks with extreme sensitivity to perturbations.

The title of the manuscript “Perfect Absorption in Chaotic Cavities with or without Hidden Symmetries” is somewhat misleading because the authors deal rather with perfect absorption in networks.

We agree that the title, and our presentation of the role of chaos, needs to be clarified in the manuscript. We have changed the title of the paper to “Perfect Absorption in *Complex Scattering Systems* with or without Hidden Symmetries”. As outlined in the Reviewer #1 and #3 comments, we have now dropped many mentions of “chaos” where they might imply a restriction to the generality and breadth of our results.

Microwave networks with broken time reversal symmetry which utilized microwave circulators were introduced in Phys. Rev. E 81, 046204 (2010). This reference should be cited additionally to the references [27] and [28].

We have added the reference as new reference [4].

Summarizing, the manuscript is well written and reports important new results. I recommend it for the publication.

We thank the reviewer for this endorsement of the paper.

Reviewer #3:

The presented manuscript reports experimental investigation on the realization of an single frequency absorber in a microwave junction network. The approach used is based on ideas from coherent perfect absorption (CPA) in complex systems. The main claim of the paper, as stated in the introduction, is to have achieved CPA in a classical chaotic structure without symmetries.

We realize from the last line of this comment that we have not properly framed the setting and generality of our results. In fact our approach does not require the presence of classical chaos, or rely on it in any way. As outlined above and below, we have modified the manuscript in numerous places to emphasize this important point.

1) The abstract of the paper focuses on biological applications, however, in the paper nothing is discussed except the realization of a single frequency absorber in a microwave system. On the theme of perfect absorption there is a vast literature in photonics made by hundreds of papers that exploits many different approaches and not limited by the CPA. Providing a work on an absorber, not even at photonics frequencies but at microwaves, without demonstrating any important application, does not certainly met the high standards of NCOMM but just deserve specialized literature.

We regret giving the impression that biological applications would be discussed in the manuscript. It is standard practice to mention the context of the work in the abstract in order to attract a broad readership. In the original manuscript we minimized discussion of all applications for the sake of brevity and used the bulk of the manuscript to directly demonstrate our main thesis. We have now removed mention of biological applications and expanded discussion of other previously mentioned applications, namely long-range wireless power transfer and sensing of subtle changes in the scattering environment, as well as new discussion of secure communications. These significant applications, and many others, follow from the generality and fundamental nature of our result. It should also be noted that the CPA condition is achieved at many frequencies, not just a single frequency, as illustrated in Fig. 2. We now do a

better job of explaining the generality of our results in the abstract and paragraphs 2, 4 (lines 99-112), 11 (lines 186-191), and the applications and conclusion paragraphs (16 and 17).

2) I do not agree the structure of the authors is chaotic. On the contrary, is a simple network made by 4 segment and there is no chaos in it. As a proof of this statement, the paper lack any sort of chaotic characterization of the system, e.g., the KS entropy, the Lyapunov exponent,... or any other metric that can demonstrate that the system is chaotic. The system used it is simply a network made by four segment.

First, we would like to remind the referee of some key wave/quantum chaos concepts. Indeed, wave chaos traces the signature of its underlying classical chaos in the statistical properties of the spectrum (and other observables) of the corresponding wave system. A typical test for the chaotic nature of a wave system is the so-called level statistics and its comparison with the Random Matrix Theory (RMT) predictions. Among them, the nearest-neighbor level spacing distribution $P(s)$ is the most typical of these measures -- demonstrating a Wigner-Dyson behavior according to RMT. The network that we have analyzed here has passed this test (and many others) in a sequence of experiments and analysis. Obviously RMT is an idealization describing what is commonly known as “hard chaos”. In typical circumstances one observes non-universal deviations from RMT (e.g. new ref. [29]) – a theme that it is at the heart of semiclassical theories.

As discussed in earlier responses, chaos is not a requirement for the efficacy of our generic CPA method. Neither are symmetries (such as time-reversal invariance) that were previously assumed by all other researchers to be necessary. This serves to illustrate the generality and significance of our results. The literature is filled with studies that either implicitly or explicitly assume requirements on the system showing CPA that are not necessary or relevant. We have endeavored in the revised manuscript to clarify the role of chaos in our results. The fourth (lines 99-112), seventh (lines 150-152), and conclusion paragraphs of the text now state that even systems that are wave chaotic can show CPA, but this is not a necessary condition. In fact many practical “chaotic” systems show significant deviations from purely chaotic behavior (i.e. they have strong periodic orbits or short orbits between the ports in their underlying classical behavior, or they have a mixed phase space of chaotic and regular behavior, etc.) Independent of these details, and even basic symmetries such as time-reversal invariance, our general CPA approach works!

3) Besides these points, I want to emphasize that this work, in a nutshell, is based on adjusting the parameters of an elementary microwave network to achieve an absorbing state at a specific frequency. That is to say, the while work is about engineering a simple filter, with no application described. In this respect, this is simple optimization work that should be published in specialized journals.

We have modified the abstract and paragraphs 2, 4, 11, and the applications and conclusion paragraphs (16 and 17) to help readers realize the broadness and generality of our results, and to see beyond the particular experimental realization presented in the paper. We also discuss in more detail specific applications enabled by our discoveries in the penultimate paragraph (16) of the main text.

4) As a minor note, I also observe a huge discrepancy between simulations and theory (Fig. 2): all predictions are different and do not follow the same evolution. It is not clear to me what the authors want to demonstrate with this analysis: it just shows a very confusing description.

As now noted in paragraph 11 of the manuscript (lines 186-191), the scattering properties of this system are complex and the exact behavior of the scattering matrix and the evolution of zeroes of the S-matrix is very difficult to capture through either an analytical model or numerical model. This is a well-established property of complex scattering systems in the semiclassical limit. Despite this difficulty, the CPA condition can still be established, demonstrating the fundamental nature of our result. Both the experiment and the numerical (approximate) treatment of the experiment show the same *generic* behavior. This is precisely what is expected from complex scattering systems, especially those that have no special symmetries, as exclusively used in the past.

In summary, I find this paper very weak. Its merit is not properly addressed, its scope is very narrow and the results described are a trivial optimization of a simple microwave network. I suggest the authors to submit this work to a more specialized paper.

We appreciate this set of remarks and have used them to restructure the presentation to make our key results clear and to emphatically articulate and delineate their generality. We have also used this opportunity to explain several important applications enabled by the results.

We feel that the reviewer's reports have greatly strengthened the paper and have guided us in correctly laying out the results and to clearly explain their fundamental significance and generality to a broad audience.

REVIEWER COMMENTS

Reviewer #1 (Remarks to the Author):

The authors answered to my points of concern, especially concerning their statement that they use a 'perfect' quantum chaotic system. They revised this statement and replace 'chaotic scattering system' by 'complex scattering systems'. However, the question remains whether their system is a 'perfect' complex scattering system. General proofs exist (The author cite the papers by Berkolaiko et al. and by Pluhar et al.) for the limit of large quantum graphs which are constructed from $N \rightarrow \infty$ vertices, where the influence of orbits on the graph, which are trapped on individual bonds seems to become negligible. This is not the case for the tetrahedral quantum graph / microwave networks used in the experiments. This system, as also stated by the third author, is not a 'generic' chaotic or complex scattering system.

I'm making this remark because in the abstract the authors state, that they use a 'complex scattering system'. For me this is confusing because in their reply to my report they state that the size of complexity or chaoticity is irrelevant. To overcome this confusion, the authors should state explicitly what kind of requirements are needed to realize a CPA. In other words, what happens if they use even less bonds or many more? These cases should also be realizable in their experiments. Furthermore, this CPA realization works at exactly one frequency. Is it possible to realize multi-frequency CPAs using microwave networks? Finally, the authors give several examples of practical applications which could be motivated by their suggestion for a CPA, but it is difficult for me to imagine how their microwave network system can be applied to these systems?

Generally, when looking at the report of the third Referee, I conclude that the title, abstract and also the central line of argumentation in the manuscript is misleading. The authors give the reader the impression that they use a complex scattering system, however, they use a system which is not perfectly complex due to the influence of these trapped orbits, caused by backscattering at the vertices.

I expected that this property poses difficulties on the realization of a CPA which relies on the destructive interference of the microwaves but the experiments show convincingly that the authors realized a CPA, and also demonstrate that they reached the CPA situation not via complete absorption through the attenuator. To get a feeling for the efficiency of the CPA it would be interesting to see P_{out}/P_{in} in a larger frequency range around the resonances shown in Figs. 3 a and 5 a. I would recommend publication after clarifying the above points by incorporating corresponding changes in the manuscript to avoid confusion.

Reviewer #2 (Remarks to the Author):

The revised manuscript is significantly improved. First of all the authors generalized their statement about the role of full chaos to achieve CPA. Now they claim that actually chaos, and especially full chaos, is not required to achieve CPA. This is an important achievement which significantly increases the applicability of the work.

The authors properly dealt with the remarks of the First Referee. Indeed, the tetrahedral graphs with time reversal invariance have some properties characteristic of wave chaotic systems and are quite well described by the Gaussian orthogonal ensemble (GOE) in the Random Matrix Theory (RMT). However, there are also some deviations, especially in higher order correlation functions from the GOE predictions. The situation is improved when one deals with the networks with broken time reversal symmetry. Such networks can be very well described by the Gaussian unitary ensembles (GUE) in RMT and the deviations from the GUE are actually not observed. One should point out that the authors in this manuscript considered both systems, with and without time reversal symmetry.

The authors also properly dealt with the opinion of the Third Referee: "I do not agree the structure of the authors is chaotic. On the contrary, is a simple network made by 4 segment and there is no chaos in it." First of all the tetrahedral network counts 6 bonds. Secondly, there are a lot of theoretical and experimental

papers which dealt with such systems, showing that their properties are quite well described by the GOE or GUE in RMT.

Summarizing, I fully recommend this revised manuscript for the publication.

Reviewer #3 (Remarks to the Author):

Second report on " Perfect Absorption in Complex Scattering Systems with or without Hidden Symmetries"

I read the authors rebuttal and the revised manuscript. While the authors did a great work in clarifying many erroneous aspects that were originally present in the manuscript, the resulting work essentially confirms all my previous points and at such I really do not see the case for NCOMM.

For example, one of my critical point was the strong discrepancy between theory and experiments, which were not addressed and conversely claimed with good agreement in the original manuscript. The author answer is essentially that the problem is difficult and that they cannot provide a better measure. This does clearly not indicate the level of exceptional findings that readers of NCOMM are expecting.

Another example, on the question of chaos, the authors tried to assemble now an unprovable answer on wave chaos. I think the authors know very little in this field, and I suggest them to study more, both in chaos as in wave chaos.

In wave chaos, the main problem is to study the link between a classically scattering system, which is described by classical equations that are nonlinear and at such can show chaos, to the quantum equivalent described by a wave equation, which is a linear object and at such does not have chaos. Many interesting properties appear in the spectra, but these are observed in classical systems that have almost infinite scattering paths (see, e.g., Ott's book on Hamiltonian chaos, or Haake in Quantum signatures of chaos), not just four segments. The authors here are also making a lot of confusions calling results of RMT. In the latter field, there are *conjectures*, and not results as the authors are incorrectly writing, that wave chaotic spectra have specific correlation properties similarly to classes of RM. However, it is well know that these are not necessary and sufficient conditions, because of the lack of any proof but just of conjectural argument. It is not possible, in general, to take a spectrum and derive results on the system using RMT, if the system is not proved to be a wave chaotic resonator with infinite scattering paths. The authors are clearly not doing that, while forcing a result that does not exist. It is also well know that second order statistical correlation tests only are itself not sufficient and higher correlation tests are required (see, e.g., Metha's book on RMT). On those there is no presence nor any discussion, clearly indicating the limited knowledge of the authors in this field and a superficial analysis provided.

This article, as I wrote in the previous report and as confirmed by the authors when answering my last remark, is an optimization of a single frequency filter. The fact that the frequency can be tuned, as the authors are erroneously arguing, does not make this system multi-frequency, it makes it a single frequency filter. I suggest the authors to check in the literature what a multi-frequency filter is, and the difference with a (tunable) single frequency filter, which is what they propose. I do not really see this measure beyond the niche application of an optimization of a single frequency filter, and at such I cannot recommend this paper for NCOMM.

Detailed reply to Referees

Reviewer #1

- The authors answered to my points of concern, especially concerning their statement that they use a 'perfect' quantum chaotic system. They revised this statement and replace 'chaotic scattering system' by 'complex scattering systems'. However, the question remains whether their system is a 'perfect' complex scattering system. General proofs exist (The author cite the papers by Berkolaiko et al. and by Pluhar et al.) for the limit of large quantum graphs which are constructed from $N \rightarrow \infty$ vertices, where the influence of orbits on the graph, which are trapped on individual bonds seems to become negligible. This is not the case for the tetrahedral quantum graph / microwave networks used in the experiments. This system, as also stated by the third author, is not a 'generic' chaotic or complex scattering system.

→ Our tetrahedral graph is not a “perfect” wave chaotic system. We have stated this in our previous response, and we have further stressed it in the revised manuscript (ms) so that there is no confusion. As the referee points out, the deviations from “universality” have been studied already in Ref [26] (and other works by these authors and by these authors together with H. Schanz), while recent experimental studies by Dietz, Sirko, and Anlage have further clarified this point (see for example new Refs. [43-45]). The origin of these deviations from the RMT predictions have been traced to the presence of short orbits which are “washed out” in the limit of large graphs (see the recent works of Pluhar and Weidenmuller and also the earlier works of Gnutzmann and Altland, and Gnutzmann, Keating, and Piotet which we cite in the revised ms in Refs. [30-32,34]).

-
- I’m making this remark because in the abstract the authors state, that they use a 'complex scattering system'. For me this is confusing because in their reply to my report they state that the size of complexity or chaoticity is irrelevant. To overcome this confusion, the authors should state explicitly what kind of requirements are needed to realize a CPA. In other words, what happens if they use even less bonds or many more? These cases should also be realizable in their experiments.

→ The use of the term “complex scattering” aimed to indicate that our setting is not “simple” (like a Fabry-Perot cavity or a periodic arrangement in photonic gratings where CPAs have been already demonstrated) and *not involving any geometric symmetries*. We would like to keep this adjective but in order to avoid unnecessary confusion, we have clarified our “nomenclature” in the revised version of the ms. We state clearly that CPA formation appears irrespective of the degree of “complexity” (two-bonds, three-bonds, or N-bonds, fully connected, or not) of the network (see the last paragraph on page 4, extending onto page 5 – lines 103-129). In the conclusion (page 17) we now state that these findings establish the validity of CPA protocols, independent of the degree of complexity of the wave transport phenomena, originating either from the influence of system-specific features in the scattering process or from the presence or the absence of an underlying classical chaotic dynamics.

-
- Furthermore, this CPA realization works at exactly one frequency. Is it possible to realize multi-frequency CPAs using microwave networks?

→ In principle, for a “fixed” network (i.e. fixed connectivity and lengths of the coaxial cables) one can find more than one CPA frequency and associated waveforms. The presence of multi-frequency CPAs occurs not only for microwave networks but also for other, more complicated, systems. To make clear this point, we have experimentally demonstrated the simultaneous appearance of two CPAs in a $\frac{1}{4}$ -bow-tie billiard (see new Fig. 5). Let us finally point out that in a recent publication (see Ref. [14]), we have provided analytical expressions, within RMT modeling, for the density of (complex) zeroes of the scattering matrix. These theoretical predictions have taken into account the number of channels and their coupling strength with the scattering set-up, and the symmetry class of the RMT ensemble. In the same publication we have also calculated the probability that a complex zero is crossing the real frequency axis (thus becoming a CPA). A non-perturbative analysis (super-symmetry) of the same quantities has been recently performed in M. Osman and Y. V. Fyodorov, Phys Rev E 102, 012202 (2020) (see added reference [48]). We have included appropriate discussion (see discussion associated with new Fig. 5 in the first full paragraph on page 12, lines 262-281) and references related to these points in the revised version of the ms.

-
- Finally, the authors give several examples of practical applications which could be motivated by their suggestion for a CPA, but it is difficult for me to imagine how their microwave network system can be applied to these systems?

→ The examples that we have presented describe situations where a microwave CPA protocol can be successfully utilized. The concepts are entirely general and apply to complex scattering systems in any form. The network implementation that we analyzed here has to be considered as a proof of principle of such CPA implementations in these more complicated set-ups. In order to provide additional evidence of the generality of the CPA principle in the microwave domain, we have performed additional experiments using a $\frac{1}{4}$ -bow-tie billiard. These new experimental results are now reported in the revised ms, thus demonstrating that the formation of CPAs is a generic phenomenon. Motivated by a comment of this referee, we have also added a new application associated with the possibility to use CPAs as reconfigurable switches. Such application is associated with the possibility to observe CPA at multiple frequencies and can be achieved via an appropriate arrangement of the relative amplitude/phase of a control wave that when injected into the CPA cavity can lead to a complete absorption of an incident signal at the same frequency (see relevant discussion at the end of page 12).

-
- Generally, when looking at the report of the third Referee, I conclude that the title, abstract and also the central line of argumentation in the manuscript is misleading. The authors give the reader the impression that they use a complex scattering system, however, they use a system which is not perfectly complex due to the influence of these trapped orbits, caused by backscattering at the vertices. I expected that this property poses difficulties on the realization of a CPA which relies on the destructive interference of the microwaves but the

experiments show convincingly that the authors realized a CPA, and also demonstrate that they reached the CPA situation not via complete absorption through the attenuator.

→ The use of the adjective “complex” in the title was/is aiming to distinguish our network platform from other “simple” experimental set-ups where e.g. geometric symmetries are present and where a CPA has been demonstrated. Instead, our platform does not have any geometric symmetries. Importantly, we have also violated time-reversal symmetry. To avoid these misunderstandings, we have explained our nomenclature in the abstract/introduction of the revised ms.

Furthermore, we agree 100% with the insightful comment of the referee that non-universal features (e.g. short orbits) that are present in our tetrahedron could, in principle, prevent the formation of CPAs by trapping the wave in parts/orbits of the network which “does not include” the lossy element. In the revised version of the ms, we strengthen further our presentation of this important point (both at the end of the introduction - page 8, lines 181-184, and in the last paragraph on page 11, extending to page 12, lines 235-261). Specifically, we now stress that non-universal features are *typical* in realistic scattering settings (e.g. bouncing ball orbits in stadium billiards) and could potentially prevent the formation of CPAs. Instead, for the case of the tetrahedron graph where short orbits associated with individual bonds are important (leading to deviations from universality), our measurements show convincingly that a CPA protocol is still successful. In fact, networks are an excellent platform to analyze the effects of such non-universal features in the formation of CPAs because they allow for a semiclassical treatment (as we have indicated at the end of the introduction of the previous version of the ms– alas without conveying clearly the message). Our hope is that the graph platform will initiate the development of a semiclassical theory of CPAs that takes into consideration these non-universal features. Such a development will be an extremely useful tool in designing efficient CPA traps.

• To get a feeling for the efficiency of the CPA it would be interesting to see P_{out}/P_{in} in a larger frequency range around the resonances shown in Figs. 3 a and 5 a.

→ We have extended the frequency range of our plots as requested by the referee and included them as insets in Figs. 3 and 6. We also supply a scale bar in Figs. 3, 5 and 6 showing the mean spacing between modes in both the microwave network and microwave billiard experimental realizations.

• I would recommend publication after clarifying the above points by incorporating corresponding changes in the manuscript to avoid confusion.

→ Thank you. We hope that in the revised ms we have addressed appropriately all the comments/questions of the referee and have provided all requested clarifications that help the reader to avoid misconceptions.

Reviewer #2

- The revised manuscript is significantly improved. First of all the authors generalized their statement about the role of full chaos to achieve CPA. Now they claim that actually chaos, and especially full chaos, is not required to achieve CPA. This is an important achievement which significantly increases the applicability of the work.

→ We thank the referee for pointing out to us that in the first version of our ms this main message had been “lost”.

-
- The authors properly dealt with the remarks of the First Referee. Indeed, the tetrahedral graphs with time reversal invariance have some properties characteristic of wave chaotic systems and are quite well described by the Gaussian orthogonal ensemble (GOE) in the Random Matrix Theory (RMT). However, there are also some deviations, especially in higher order correlation functions from the GOE predictions. The situation is improved when one deals with the networks with broken time reversal symmetry. Such networks can be very well described by the Gaussian unitary ensembles (GUE) in RMT and the deviations from the GUE are actually not observed. One should point out that the authors in this manuscript considered both systems, with and without time reversal symmetry.

→ Agreed.

-
- The authors also properly dealt with the opinion of the Third Referee: “I do not agree the structure of the authors is chaotic. On the contrary, is a simple network made by 4 segment and there is no chaos in it. “First of all, the tetrahedral network counts 6 bonds. Secondly, there are a lot of theoretical and experimental papers which dealt with such systems, showing that their properties are quite well described by the GOE or GUE in RMT.

→ We agree with the referee.

-
- Summarizing, I fully recommend this revised manuscript for the publication.

→ Thank you.

Reviewer #3

- For example, one of my critical points was the strong discrepancy between theory and experiments, which were not addressed and conversely claimed with good agreement in the original manuscript. The author answer is essentially that the problem is difficult and that they cannot provide a better measure. This does clearly not indicate the level of exceptional findings that readers of NCOMM are expecting.

→ The discrepancy between the modeling/numerical simulations and experimental data was associated with the fact that our modeling did not take into consideration the frequency dependence of the vertex scattering matrices (T-junctions). This simplification leads to quantitative differences on the exact evaluation of the CPA frequencies. We have improved our modeling by directly measuring the frequency-dependent S-matrix of each component, and then assembling their responses into a graph topology to model the full response of the network. There is now excellent agreement between theory/numerics and experiment, as evident in Figs. 2, 3 and 6. Figure S3 was also added to further demonstrate the degree of agreement between data and simulation. Of course, there are still various undetermined factors (residual absorption and scattering at each specific bond/junction, etc) which cannot be accounted exactly in the modeling. But this is exactly the reason that we have performed the experiment i.e. to demonstrate that the formation of a CPA is not an outcome of an “ideal modeling” but can be implemented in realistic (complicated) settings!

- Another example, on the question of chaos, the authors tried to assemble now an unprovable answer on wave chaos. I think the authors know very little in this field, and I suggest them to study more, both in chaos as in wave chaos. In wave chaos, the main problem is to study the link between a classically scattering system, which is described by classical equations that are nonlinear and at such can show chaos, to the quantum equivalent described by a wave equation, which is a linear object and at such does not have chaos. Many interesting properties appear in the spectra, but these are observed in classical systems that have almost infinite scattering paths (see, e.g., Ott’s book on Hamiltonian chaos, or Haake in *Quantum Signatures of Chaos*), not just four segments

→ We are afraid that the referee is taking the discussion to a direction which is completely unrelated to the message of the paper. We re-iterate what we wrote previously: The main message of this work is that CPAs occur in the absence of any (geometric or hidden) symmetries and despite the presence of non-universal features, which are typically present in many realistic scattering set-ups (on the potential importance of non-universal features in the formation of CPAs, please also see the insightful comment of referee #1 with which we agree 100%). RMT statistics, “hard chaos”, “soft chaos”, etc. are irrelevant and does not affect the formation of CPAs. Of course, in the case of “hard chaos” one might end up with some universal statistical behavior (see papers [13,14,48]) of CPAs, but again this is not the theme of the current paper. Our study asks, and answers experimentally, the most fundamental question of the validity of CPA protocols in the absence of symmetries. Importantly, our work shows that the previous interpretation of CPA as a time-reversal of a laser is rather restrictive and CPA is a broad concept. In our revised ms we have (hopefully) made this point crystal-clear.

Coming back to the essence of the referee’s comment above (which again is loosely related to the subject of this paper): There is a “common” confusion between geometric and dynamical complexity. It is a fact that chaos (chaotic scattering) is associated with an exponential proliferation of the number of periodic orbits (scattering trajectories) with increasing path-lengths. But for such a property one does not require the system to possess any geometric complexity; only dynamical complexity is necessary! After all, chaotic scattering was initially introduced in the framework of a (geometrically) simple system of three discs (see early works

of Gaspard and Rice). Fully connected graphs possess the feature of exponential proliferation of periodic orbits (PO) with increasing “length”. In particular for a fully connected graph one has an exponential proliferation of n-PO with topological entropy $\sim \log(v)$ where v is the valency of the graph ($v = 3$ in our case). But again, all of this is established knowledge (please see e.g. the original study [26], or the very nice review [28] and the more recent papers on conditions for universality on graphs [30-32]) – although we emphasize again that they are not really relevant to our study.

- The authors here are also making a lot of confusions calling results of RMT. In the latter field, there are *conjectures*, and not results as the authors are incorrectly writing, that wave chaotic spectra have specific correlation properties similarly to classes of RM. However, it is well known that these are not necessary and sufficient conditions, because of the lack of any proof but just of conjectural argument.

→ We stress once more that this paper is NOT about wave chaos as we have made clear in our previous reply (and hopefully with the revisions of the current version)! Neither is it about establishing graphs as a model of wave chaos. This has been done a long time ago by one of the authors of this paper (in a sequence of joint publications with Profs. U. Smilansky and H. Schanz) and via subsequent studies of Weidenmuller, Haake, Gnutzmann, Sirko, Tanner, Altland, Dietz etc. Deviations of the spectral and wavefunction statistics from RMT predictions have been pointed out, their origin has been identified (based on short orbit analysis), and criteria for agreement with RMT have been discussed. A very nice summary of many of these results can be found at the book of Haake; for the latter reference please use the most recent 4th edition (2018) of the book where the author dedicated chapter 7 on the study of quantum graphs. In this respect there is nothing to discuss further.

At the same time, we maintain one of the novelties of our network platform (please note that by “platform” we refer to the whole family of networks with a number of bonds which might be larger than $v = 4$): It is one of the very few systems that we are aware of where semiclassical tools can be implemented successfully, unveiling the importance of system-specific characteristics (like the presence of short orbits that lead to statistical deviations from RMT). We hope that the semiclassical toolbox will be used successfully in the analysis of CPA formation. Currently we are working along this direction.

- It is not possible, in general, to take a spectrum and derive results on the system using RMT, if the system is not proved to be a wave chaotic resonator with infinite scattering paths. The authors are clearly not doing that, while forcing a result that does not exist. It is also well known that second order statistical correlation tests only are itself not sufficient and higher correlation tests are required (see, e.g., Metha’s book on RMT). On those there is no presence nor any discussion, clearly indicating the limited knowledge of the authors in this field and a superficial analysis provided.

→ Please see our previous response.

-
- This article, as I wrote in the previous report and as confirmed by the authors when answering my last remark, is an optimization of a single frequency filter. The fact that the frequency can be tuned, as the authors are erroneously arguing, does not make this system multi-frequency, it makes it a single frequency filter. I suggest the authors to check in the literature what a multi-frequency filter is, and the difference with a (tunable) single frequency filter, which is what they propose. I do not really see this measure beyond the niche application of an optimization of a single frequency filter, and at such I cannot recommend this paper for NCOMM.

→ This is an excellent opportunity for us to help the referee appreciate the basic principle of CPA. The CPA should not be confused with a zero of the transfer function (S_{21}) of a linear filter. As opposed to a standard frequency filter, a CPA requires two (or more) ports from where you have to inject incoming waves with appropriate phases and amplitudes. In other words, a CPA is not only an “optimal” filter, but it is a device with which you induce/control signal elimination via other signals which when appropriately tailored lead to destructive interferences that trap the signal inside the weakly lossy cavity leading to their complete elimination via absorption. Finally, it should also be noted that we now include experimental demonstration of multi-frequency CPA in a fully ray-chaotic microwave billiard in the revised manuscript (see new Fig. 5).

REVIEWERS' COMMENTS

Reviewer #1 (Remarks to the Author):

The authors addressed all my concerns appropriately. They clarify what they refer to by 'complex scattering system'. Above all, they improved their manuscript considerably and added new / revised results in reply to my remarks / questions / suggestions. In its present form it definitely deserves publication in Nature Communications.

Response to the Reviewers

Reviewer #1

The authors addressed all my concerns appropriately. They clarify what they refer to by 'complex scattering system'. Above all, they improved their manuscript considerably and added new / revised results in reply to my remarks / questions / suggestions. In its present form it definitely deserves publication in Nature Communications.

We thank the referee for all his/her valuable advice on our manuscript during the review process. Driven by all these comments, we have greatly strengthened the paper with new experiments and explanations. We acknowledge the referee in guiding us to correctly lay out the results and offering us the opportunity to clearly explain the fundamental significance and generality of our paper to a broad audience.